# Genetically modified organisms and sustainable development goals: A survey of Taiwanese public opinion

Shin-Cheng Yeh[1], Pei-Xuan Chen[1], Chih-Cheng Lin[2]*, I-Chun Chiang[3]

1 Graduate Institute of Sustainability Management and Environmental Education, National Taiwan Normal University, Taipei, Taiwan, 2 Department of Fisheries Technology and Management, Center for Marine Affairs Studies, National Kaohsiung University of Science and Technology, Kaohsiung, Taiwan, 3 Department of Business Computing, National Kaohsiung University of Science and Technology, Kaohsiung, Taiwan

* cclin108@nkust.edu.tw

## Abstract

This study investigates public opinion of genetically modified organisms (GMOs), the role of GMOs in achieving the Sustainable Development Goals (SDGs), and economic preferences in Taiwan. A survey of 977 Taiwanese adults assessed their knowledge, attitudes, and willingness to pay (WTP) for non-GMO and organic products. The results show that while awareness of GMOs is high, there is a significant gap in understanding, as many respondents struggled with basic genetic concepts. Public attitudes are generally neutral, with moderate concerns about health and environmental risks. Notably, males and those with a science background exhibit more favorable attitudes towards GMO. The WTP analysis reveals a stronger preference for paying premiums on organic products over non-GMO items, indicating a higher perceived value of organic agriculture. Public perception of GMOs' contributions to SDGs is largely positive, particularly for enhancing food security (SDG 2) and alleviating poverty (SDG 1), though concerns remain regarding environmental sustainability (SDGs 14 and 15). These findings underscore the need for targeted educational initiatives and effective communication strategies to close knowledge gaps and build public trust in GMO regulation, which are essential for informed public discourse and maximizing GMOs' potential in sustainable development.

## 1. Introduction

In the face of complex global challenges such as population growth, climate change, and limited natural resources, the possible contributions of genetically modified organisms (GMOs) to food security have been discussed because of its potential to enhance crop productivity, increase resistance to pests and diseases [1], and even bolster nutritional content [2]. GMO is defined as an organism resulting from the

**Data availability statement:** All relevant data are within the paper and its Supporting information files.

**Funding:** The author(s) received no specific funding for this work.

**Competing interests:** The authors have declared that no competing interests exist.

transfer of a DNA segment (gene) from one living entity (such as plants, animals, or microorganisms) to another, typically accomplished through recombinant DNA techniques [3]. The application of GMOs in agriculture has been demonstrated to have a positive impact on farmers' incomes, economic access to food, and increased crop tolerance to various biotic and abiotic stresses [4]. Furthermore, GMOs have the potential to address food security challenges by enhancing crop varieties, farming practices, and the utilization of fertilizers and agrochemicals. As such, GMOs are also considered to be beneficial in achieving the Sustainable Development Goals (SDGs), a set of global objectives adopted by the United Nations in 2015 to address social, economic, and environmental challenges and promote sustainable development [5].

The potential contributions of GMOs to the achievement of the SDGs have been explored across a broad range of SDGs [6]. Tyczewska et al. [7] have reviewed the potential applications of GMOs in areas such as food security, land conservation, waste reduction, reduced carbon emissions, water preservation, and carbon sequestration. They concluded that GM (genetically modified, GM) crops have the potential to reduce both pesticide use and soil damage, as well as decrease greenhouse gas (GHG) emissions. Moreover, according to Abdul Aziz et al. [8], GM technology has the potential to contribute to the achievement of SDG 13 – climate action and SDG 15 – life on land, by promoting more environmentally sustainable agriculture. Paarlberg & Smyth [9] have even suggested that GM technology presents a significant new opportunity for crop improvement, with the potential to benefit farmers and consumers, particularly in low- and middle-income countries. However, while GMOs could advance SDGs, there are also potential risks associated with their use.

Despite the potential benefits of GMOs, their development and deployment remain contentious. Public apprehension frequently centers on perceived health risks, environmental repercussions, and ethical concerns—issues often exacerbated by misinformation and media sensationalism [10]. In Taiwan, as in many other regions, persistent skepticism regarding GMO safety and a prevailing distrust in regulatory authorities continue to influence public attitudes [11,12]. These concerns underscore the importance of examining not only the scientific and economic dimensions of GMOs but also the sociocultural and regulatory contexts that shape public acceptance. As Nasser et al. [13] emphasize, the adoption of GM technology poses several challenges that may hinder progress toward achieving the SDGs. Specifically, concerns about allergens and toxins, antibiotic resistance, and a lack of long-term research could hinder progress towards SDG 3 – good health and well-being. Additionally, issues such as insecticide resistance, loss of biodiversity, and safety and regulatory concerns could impede progress towards SDG 15 – life on land. Furthermore, conspiracies or myths revolving around GMOs have yet to be eradicated. For instance, some individuals may believe that GMOs are part of a plot by corporations to control the global food supply or to depopulate the world by causing diseases [10]. Several debates concerning GMOs revolve around notions of unnaturalness, lack of trustworthiness, moral quandaries, uncertainties, potential health risks, and associated hazards [11]. However, it is crucial to underscore that these assertions lack substantiation from empirical scientific investigations. As a result, Erokhin & Komendantova [10] have argued that the dissemination

of misinformation and conspiratorial narratives concerning GMOs bears noteworthy ramifications for public policy, scientific inquiry, and overall public health. The propagation of such misinformation and conspiracy theories has the potential to undermine public perception and amplify reservations toward the consumption of GMOs.

Public opinion has a significant impact on the development and marketing of GMOs [14]. Understanding how public opinions on GMOs are formed is essential due to the direct causal relationship between public opinion and public policy [15]. According to past studies, public knowledge and awareness of GMOs are crucial in shaping consumers' views on the benefits and risks associated with GM foods [16,17]. Furthermore, individuals' familiarity with GMOs impacts their willingness to consume GM products [18]. Therefore, the aspects of public opinion towards GMOs investigated in this study include knowledge, attitude, and willingness to pay (WTP).

It is essential to identify current attitudes toward biotechnology in different countries to understand public fears and determine potential knowledge gaps [19]. Taiwan, as a developed country in East Asia, has shown through past research that its citizens support the development of GM technology because it is a global trend [20]. However, Chen [11] found that the public's willingness to consume GM foods was not high, with a mean score of 3.16 out of 7. Therefore, there is a need to investigate public opinion regarding GMOs in Taiwan. Moreover, while public perceptions of GMOs have been extensively studied in Western contexts—particularly concerning the preference for "natural" food and the perceived unnaturalness of GMOs [21]—there is limited research comparing willingness to pay (WTP) for GM, non-GM, and organic food in the East Asian context. Likewise, although GMOs have been framed as contributors to the SDGs in prior literature [7,8], few studies have explored public perceptions linking GMOs and SDGs in East Asia The research purposes of this study include: first, investigating the public's knowledge, attitudes, and WTP concerning GMOs. Second, we analyze the public's perceived relationship between GMOs and the SDGs. This study offers valuable insights into the complex dynamics of public opinion on GMOs and their potential impact on sustainable development initiatives. By focusing on the Taiwanese context, we provide new empirical evidence on how GMOs are perceived in relation to sustainable development, addressing both regional and thematic gaps in the existing literature.

## 2. Materials and methods

### 2.1 Questionnaire design

As the primary objective of this study is to understand the current perceptions of GMOs among the Taiwanese public. Additionally, this research investigates the relationship between GMOs and sustainable development goals, as well as the WTP for non-GMO products. Data were collected through a questionnaire survey and subsequently analyzed using SPSS version 23.

**2.1.1 Questionnaire design for the knowledge dimension.** This section of the questionnaire includes subjective knowledge and objective knowledge. To understand the public's basic awareness of GMOs, the survey first asks respondents whether they have heard of terms related to GMOs. Based on a literature review, public knowledge related to GMOs can be divided into two categories: subjective knowledge, referring to the perceived understanding of GMOs, and objective knowledge, which measures the actual understanding of GMOs [21]. On the scale of subjective knowledge, respondents are asked about their self-rated perceived level of understanding of GMOs. In contrast, the scale of objective knowledge tests respondents with true or false question questions that have correct answers. These questions are based on middle school-level biological and GMO-related knowledge.

**2.1.2 Questionnaire design for the attitude dimension.** The attitude dimension of the questionnaire, drawing on the findings of Scott et al. [21], includes four sub-dimensions: general attitude, perceived risks, perceived benefits, and trust. The scales in this dimension mostly employ a five-point Likert scale, ranging from strongly disagree to strongly agree, to evaluate public attitudes towards GMOs.

To construct the perceived risks scale, items were informed by common concerns identified in previous studies, including health, environmental, and socio-economic factors [5]. Items in the perceived benefits scale reflected findings that GMOs may increase crop yields, reduce production costs, and improve food quality [22]. Trust-related items were

developed based on the work of Scott et al. [21], and Zhang et al. [20,23], focusing on trust in government and scientific sources.

**2.1.3 Questionnaire design for the WTP.** WTP refers to the maximum price individuals are willing to pay for a product or service, reflecting the value they place on the item [24]. This study measured willingness to pay a premium for non-GMO and organic products. The questionnaire design was informed by previous research [25,26], with responses categorized into percentage-based intervals (e.g., 0%, 1%–10%, 11%–20%, etc.) to estimate WTP.

**2.1.4 Questionnaire design for the relationship between GMOs and SDGs.** This section of the questionnaire assessed public perceptions regarding the relationship between GMOs and the SDGs. Respondents were first asked about their level of understanding of the SDGs and whether they had encountered any related images. We also assessed respondents' self-perceived knowledge of the SDGs to better elucidate their basic understanding. The questionnaire then presented the 17 SDGs, accompanied by a five-point scale with the following options: very negative (−2), negative (−1), neutral (0), positive (+1), and very positive (+2). This scale enabled respondents to indicate their perceptions of the impact of GMOs on each of the 17 SDGs.

## 2.2 The research subjects and data collection

The research subjects were Taiwanese adults over 18 years old. This study utilized Google Forms to create the questionnaire and employed two distribution methods. The first method involved sending the questionnaire through personal networks, asking friends and family members to forward it to others to expand the sample size. The second method involved distributing the questionnaire through the internet. Since there is no requirement for mandatory IRB approval to conduct public opinion surveys in Taiwan, we followed the ethical guidelines set forth by the Taiwanese Sociological Association [27]. These guidelines emphasize three key principles: (1) preventing conflicts of interest by upholding professional standards, transparently disclosing research funding sources, and avoiding any personal gain that might compromise the research; (2) maintaining confidentiality by protecting participants' rights and implementing a plan to secure their information; and (3) obtaining informed consent by verbally informing participants of their rights and clarifying the scope of information disclosure. Respondents were informed that they would participate in the survey online and that all data would be de-identified and reported only in aggregate form. In addition, a consent statement was included on the introduction page of the Google Form, explaining the purpose of the study, data anonymization, and the intended use of results for academic publication. Proceeding to complete the survey was taken as confirmation of participants' consent to participate and for their anonymized responses to be used in this study. The recruitment period for this study spanned from March 1 to May 31, 2021. A total of 977 valid responses were collected, and the demographic distribution of respondents is presented in Table 1.

## 3. Results

### 3.1 Public knowledge on GMOs

Fig 1 presents public responses to the question on the awareness of GMO-related terms. It shows that only 32 people, accounting for 3.3%, were completely unaware of GMO-related terms. Those with a vague awareness numbered 217, making up 22.2%. There were 168 people who had a clear impression of GMOs, constituting 17.2%. Frequently hearing GMO-related terms were 311 people, representing 31.8%, while 249 people, or 25.5%, reported always hearing GMO-related terms. These results indicate that the majority of the public in Taiwan has heard of GMO-related terms.

Fig 2 presents public responses to questions on their self-rated perceived level of understanding of GMOs. It shows that 92 people, accounting for 9.4%, are completely unfamiliar with GMOs. Those who are slightly familiar number 158, making up 16.2%. A total of 641 people, or 65.6%, are somewhat familiar with GMOs, which is the highest among all options. There are 71 people, representing 7.3%, who are fairly familiar with GMOs. Only 15 people, or 1.5%, are very

**Table 1. Demographic distribution of respondents.**

| Gender | Number | Percentage |
|---|---|---|
| Male | 424 | 43.4 |
| Female | 553 | 56.6 |
| **Age** | **Number** | **Percentage** |
| Under 20 | 78 | 8 |
| 20-29 | 399 | 40.8 |
| 30-39 | 206 | 21.1 |
| 40-49 | 142 | 14.5 |
| 50-59 | 110 | 11.3 |
| Over 60 | 42 | 4.3 |
| **Education** | **Number** | **Percentage** |
| Junior college student | 17 | 1.7 |
| Junior college graduate | 53 | 5.4 |
| University student | 286 | 29.3 |
| University graduate | 190 | 19.4 |
| Graduate school student | 161 | 16.5 |
| Graduate school graduate | 237 | 24.3 |
| Other | 33 | 3.4 |
| **Education field** | **Number** | **Percentage** |
| Science | 411 | 42.1 |
| Humanities and Social Science | 532 | 54.4 |
| Other | 34 | 3.5 |

familiar with GMOs. These results indicate that the majority of the public believes they have only a slight or somewhat degree of subjective knowledge about GMOs.

Table 2 presents public responses to objective knowledge questions. It shows that among the five questions in the objective knowledge section, only the second question, "Gene transfer does not occur naturally," and the fourth question, "Men and women have different numbers of chromosomes," had correct response rates exceeding 50%. This indicates a significant lack of objective knowledge about GMOs among the public in Taiwan.

### 3.2 Public attitude towards GMOs

Table 3 presents the general attitude towards GMOs. Each level of agreement was scored from 1 (strongly disagree) to 5 (strongly agree). For negatively worded statements—such as "GMOs violate the laws of nature," "GMOs violate moral principles," and "GMOs violate religious beliefs"—responses were reverse-coded. This means that a response of 5 (strongly agree) was recoded as 1, and 1 (strongly disagree) as 5. Reverse scoring was used to ensure that across all items, higher scores consistently reflected a more favorable attitude toward GMOs. Overall, the public in Taiwan tends to hold a neutral attitude towards GMOs, with the average mean score being 3.02.

Table 4 presents the perceived risks of respondents towards GMOs. Each level of agreement was scored from 1 to 5. Overall, the public in Taiwan tends to hold slightly below average perceived risks towards GMOs, with the average mean score being 2.78.

Table 5 presents the perceived benefits of respondents towards GMOs. Each level of agreement was scored from 1 to 5. Overall, the perceived benefits of the public towards GMOs in Taiwan are above average, with the average mean score being 3.52.

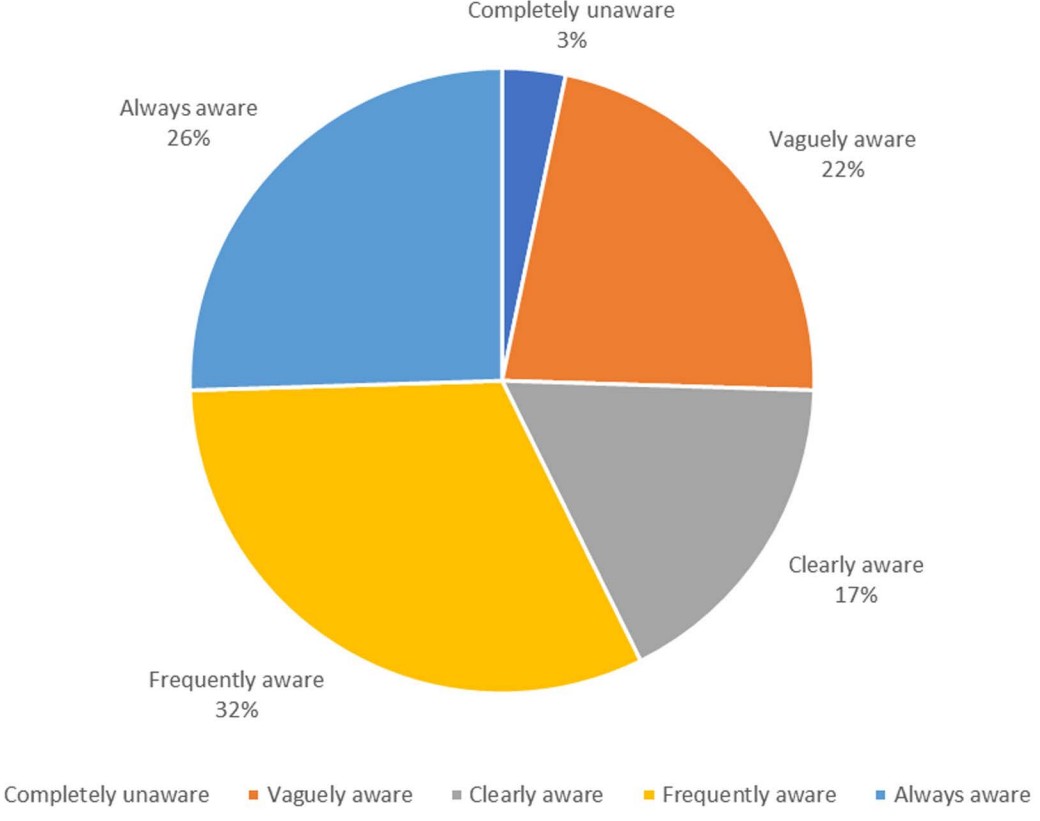

**Fig 1. Frequency distribution of awareness of GMO-related terms.**

Table 6 presents the trust of respondents towards GMOs. Each level of agreement was scored from 1 to 5 for the statements "GMOs can be properly regulated by technology" and "I have confidence in the government's management of GMOs." Two negatively worded items statements "The development of GMO technology will exceed human control" and "The public should participate in the management of GMOs" were reverse-coded. With an average mean score of 2.56, the results indicate that public trust in the management of GMOs is insufficient.

### 3.3 Public WTP towards non-GMO and organic food

Fig 3 presents the public willingness to pay a premium for non-GM food compared to GM food. The distribution of the public's willingness to pay a premium for non-GM food mostly falls below 31–40%. The highest proportion is those willing to pay an additional 11–20%, with 234 people, accounting for 24% of the respondents. This is followed by 197 people, or 20.2%, who are willing to pay an additional 1–10%. Next, 167 people, or 17.1%, are willing to pay an additional 21–30%.

Fig 4 presents the public willingness to pay a premium for organic food compared to conventional food. The distribution of the public's willingness to pay a premium for organic food mostly falls below 31–40%. The highest proportion of respondents is willing to pay an additional 11–20%, with 215 people, accounting for 22% of the respondents. This is followed by 207 people, or 21.2%, who are willing to pay an additional 21–30%. Next, 139 people, or 14.2%, are willing to pay an additional 1–10%.

To accurately determine the actual WTP value, this study employs a weighted statistical analysis. The formula calculates the weighted mean ($\bar{x}$), where $x$ denotes the midpoint of each interval (e.g., the midpoint for the 11% to 20% interval

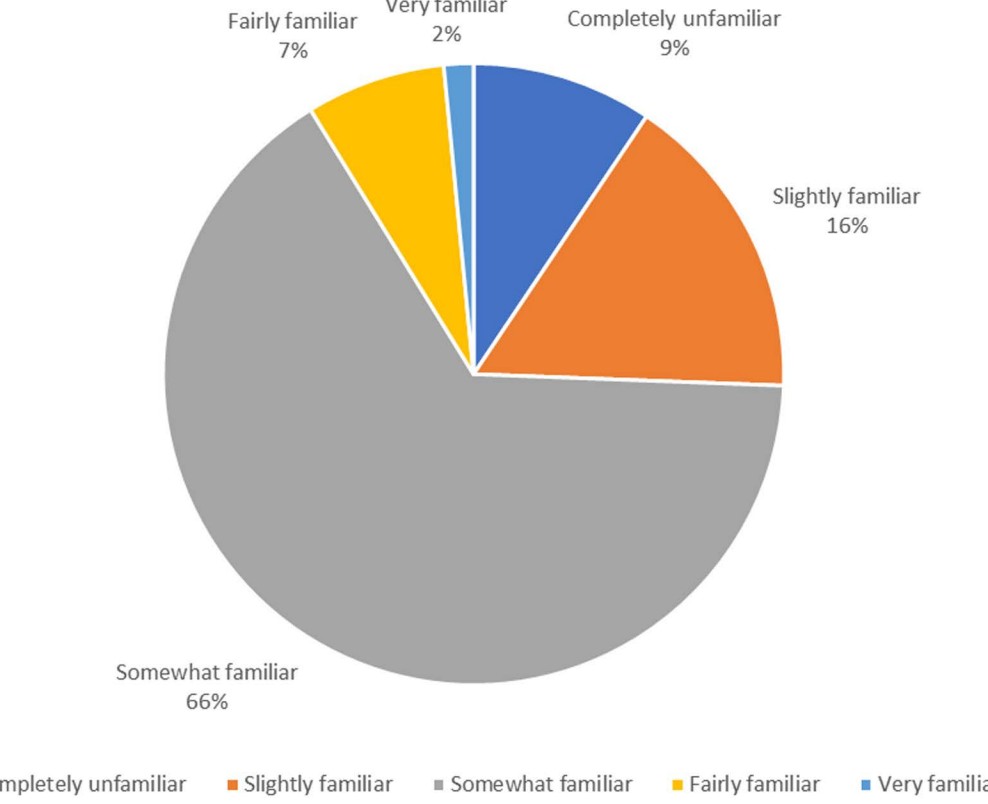

**Fig 2. Frequency distribution of public perceived understanding of GMOs.**

**Table 2. Frequency distribution of responses to objective knowledge questions.**

| Question | Correct/Incorrect | Number of People | Percentage |
|---|---|---|---|
| Genes are made up of chromosomes | Correct | 144 | 14.7% |
| | Incorrect | 833 | 85.3% |
| Gene transfer does not occur naturally | Correct | 550 | 56.3% |
| | Incorrect | 427 | 43.7% |
| More DNA means a higher organism | Correct | 365 | 37.4% |
| | Incorrect | 612 | 62.6% |
| Men and women have different numbers of chromosomes | Correct | 519 | 53.1% |
| | Incorrect | 458 | 46.9% |
| Consuming GMO products can alter the consumer's genes | Correct | 349 | 35.7% |
| | Incorrect | 628 | 64.3% |

is 15.5%), and *f* represents the frequency or number of individuals in each interval, and n is the total number of individuals By multiplying the midpoint of each interval by the corresponding frequency, summing these products, and then dividing by the total number of individuals, the weighted mean value is obtained. This approach ensures that the WTP value accurately reflects the distribution of respondents' willingness to pay across different intervals. The formula is as follows:

**Table 3. Responses on general attitude towards GMOs.**

| Statement | Do you agree with the statement? | | | | | Mean |
|---|---|---|---|---|---|---|
| | Strongly disagree | Disagree | Neutral | Agree | Strongly agree | |
| GMOs violate the laws of nature | 2.6% | 8.7% | 46.4% | 36.1% | 6.2% | 2.65 |
| GMOs violate moral principles | 4.1% | 17.2% | 57.6% | 18.4% | 2.7% | 3.02 |
| GMOs violate religious beliefs | 10.7% | 23.5% | 53.3% | 10.8% | 1.5% | 3.31 |
| GMOs contribute to human development | 2.6% | 11.2% | 62.2% | 20.5% | 3.6% | 3.12 |
| Average Mean | | | | | | 3.02 |

**Table 4. Responses on perceived risks towards GMOs.**

| Statement | Do you agree with the statement? | | | | | Mean |
|---|---|---|---|---|---|---|
| | Strongly disagree | Disagree | Neutral | Agree | Strongly agree | |
| GMOs have many unknown side effects | 9.3% | 33.6% | 35.2% | 18% | 3.9% | 2.74 |
| GMOs have negative impacts on adult health | 3.3% | 23.6% | 63.6% | 8.2% | 1.3% | 2.81 |
| GMOs have negative impacts on children's health | 6.2% | 24.1% | 55.8% | 11.2% | 2.8% | 2.81 |
| GMOs have negative impacts on the natural ecosystem | 7.5% | 30.7% | 43% | 15% | 3.8% | 2.77 |
| Average Mean | | | | | | 2.78 |

**Table 5. Responses on perceived benefits towards GMOs.**

| Statement | Do you agree with the statement? | | | | | Mean |
|---|---|---|---|---|---|---|
| | Strongly disagree | Disagree | Neutral | Agree | Strongly agree | |
| GMOs can increase global food production | 1.4% | 3.6% | 27.8% | 53.6% | 13.5% | 3.74 |
| GMOs can stabilize food prices | 2.3% | 7.7% | 33.5% | 46.1% | 10.5% | 3.55 |
| GMOs can reduce pesticide usage | 1.9% | 7.2% | 30.8% | 49% | 11.1% | 3.60 |
| GMOs can serve as "edible vaccines" to prevent diseases | 3.7% | 14.7% | 46.3% | 29.8% | 5.5% | 3.19 |
| Average Mean | | | | | | 3.52 |

**Table 6. Responses on public trust towards GMOs.**

| Statement | Do you agree with the statement? | | | | | Mean |
|---|---|---|---|---|---|---|
| | Strongly disagree | Disagree | Neutral | Agree | Strongly agree | |
| GMOs can be properly regulated by technology | 6.7% | 29.2% | 43.4% | 18.4% | 2.4% | 2.81 |
| I have confidence in the government's management of GMOs | 12.6% | 27.7% | 47.7% | 10.8% | 1.1% | 2.60 |
| The development of GMO technology will exceed human control | 1.4% | 6.7% | 38.7% | 43.6% | 9.6% | 2.47 |
| The public should participate in the management of GMOs | 1.8% | 5.9% | 32.4% | 47.5% | 12.3% | 2.37 |
| Average Mean | | | | | | 2.56 |

$$\overline{x} = \frac{x_1 f_1 + x_2 f_2 + \cdots + x_n f_n}{f_1 + f_2 + \ldots + f_n}$$

By applying the respondents' WTP data for non-GM food to the weighted formula, we obtain a value of 25.4%. This indicates that the respondents are willing to pay approximately 25.4% more to purchase non-GM food. Similarly, when applying the respondents' WTP data for organic food to the weighted formula, we obtain a value of 30.12%. This suggests that

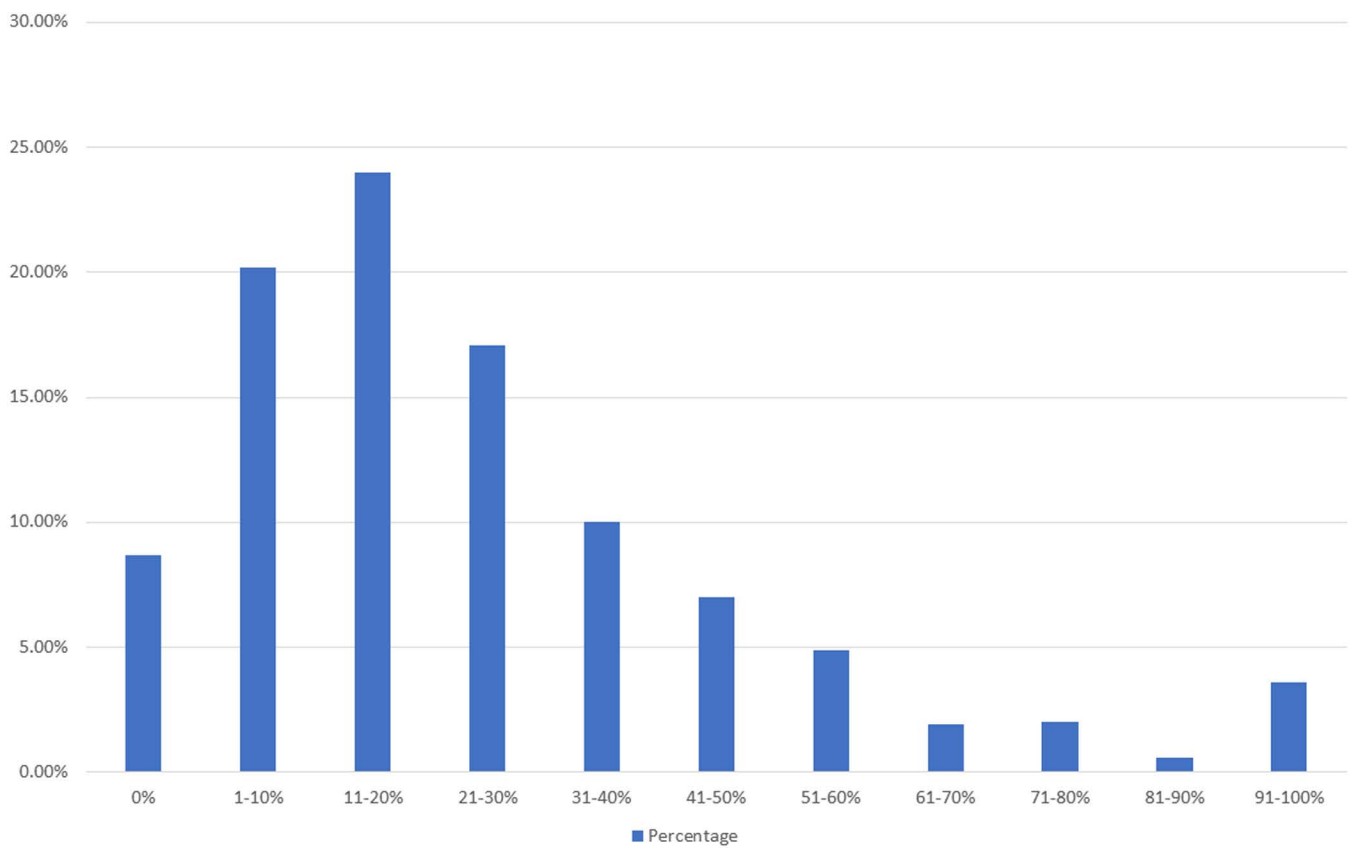

**Fig 3. Distribution of public willingness to pay a premium for non-GM food.**

the respondents are willing to pay approximately 30.12% more to purchase organic food. Thus, the public is more inclined to pay a higher premium for organic products compared to non-GM food.

### 3.4 Public perception of the relationship between GMOs and SDGs

Table 7 presents the public perception in Taiwan of the relationship between GMOs and SDGs. It shows that 46.1% of the public believes there is a positive relationship between GMOs and SDG 2 – zero hunger, with 15.7% considering this relationship to be very positive. This makes SDG 2 the goal most positively associated with GMOs among all the SDGs. Additionally, 36.9% of the public perceives a positive relationship between GMOs and SDG 1 – no poverty, with 7.4% viewing this relationship as very positive, ranking SDG 1 second in positive association with GMOs. The third-ranking goal is SDG 9 – industry, innovation, and infrastructure, with 36.9% of the public seeing a positive relationship with GMOs, and 5.6% considering it to be very positive.

Regarding negative relationships between GMOs and SDGs, 25.9% of the public believes that GMOs have the most negative relationship with SDG 15 – life on land. This is followed by SDG 14 – life below water, with 24.3% of respondents perceiving a negative impact, and SDG 12 – responsible consumption and production, with 20.9% seeing GMOs as having a negative influence. Additionally, SDG 5 – gender equality is the only goal where the negative impact is perceived to outweigh the positive influence, with 19.4% of respondents believing that GMOs negatively affect SDG 5, compared

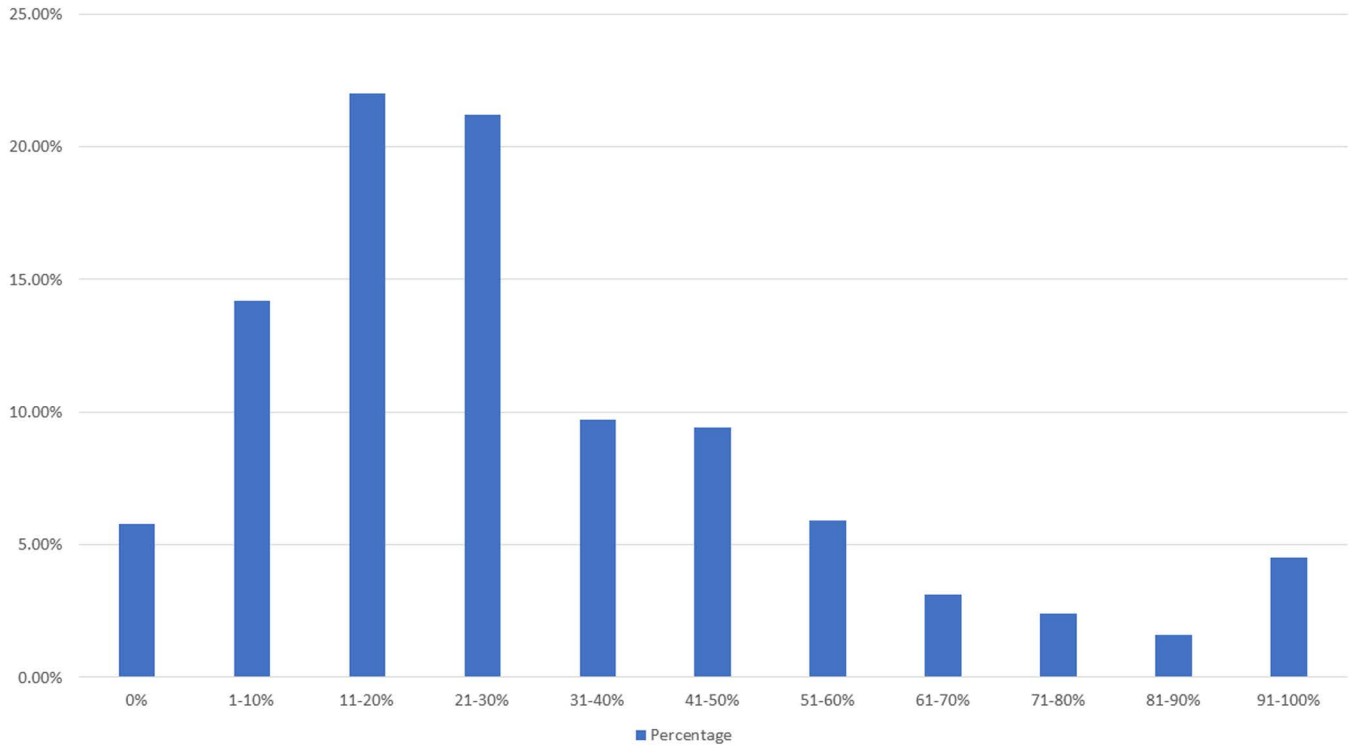

**Fig 4. Distribution of public willingness to pay a premium for organic food.**

**Table 7. Responses on perception of the relationship between GMOs and SDGs.**

| SDG | What do you think about the relationship between GMOs and SDG? | | | | |
|---|---|---|---|---|---|
| | Very positive | Positive | Neutral | Negative | Very negative |
| SDG 1 | 7.4% | 36.9% | 41.9% | 11.1% | 2.8% |
| SDG 2 | 15.7% | 46.1% | 30.5% | 6.4% | 1.3% |
| SDG 3 | 5.4% | 26.7% | 51% | 13.5% | 3.4% |
| SDG 4 | 2.8% | 18.1% | 62.1% | 13.2% | 3.8% |
| SDG 5 | 2.7% | 9.8% | 68.1% | 13.7% | 5.7% |
| SDG 6 | 4.8% | 29.3% | 50.3% | 12.4% | 3.3% |
| SDG 7 | 5.3% | 29.7% | 50.2% | 11.5% | 3.4% |
| SDG 8 | 5.4% | 34% | 47.5% | 10.8% | 2.3% |
| SDG 9 | 5.6% | 36.9% | 47.2% | 8.5% | 1.7% |
| SDG 10 | 4% | 24.2% | 53.9% | 13.4% | 4.5% |
| SDG 11 | 4.6% | 26.3% | 52.9% | 12.5% | 3.7% |
| SDG 12 | 5.2% | 28.9% | 45% | 16.1% | 4.8% |
| SDG 13 | 5.7% | 28.9% | 47.2% | 13.8% | 4.4% |
| SDG 14 | 4.8% | 22% | 48.8% | 18.4% | 5.9% |
| SDG 15 | 5% | 22.4% | 46.7% | 20.0% | 5.9% |
| SDG 16 | 3.8% | 19% | 56.9% | 15.3% | 5% |
| SDG 17 | 4.2% | 19.4% | 59% | 13.3% | 4.1% |

to 12.5% who see positive benefits. SDG 5 also has the lowest perceived relevance to GMOs, with 68.1% of the public holding a neutral view on the relationship between GMOs and SDG 5.

### 3.5  Cross-analysis of the variables

This study conducts a cross-analysis of the research variables. Initially, independent sample t-tests and ANOVA are used to examine the differences in opinions on GMOs across various personal background variables. Following this, the correlations among public knowledge, attitudes, and willingness to pay for GMOs are analyzed.

On the public knowledge, ANOVA analysis results indicate significant differences in subjective knowledge across different age groups ($p < 0.001$). Individuals under 20 years old have higher subjective knowledge scores compared to those aged 30–39 and 60 and above, while those aged 20–29 score higher in subjective knowledge than those aged 30–39, 40–49, and 60 and above. Additionally, there are significant differences in objective knowledge across age groups ($p < 0.005$), with individuals aged 20–29 scoring higher in objective knowledge than those aged 60 and above.

The study found significant differences between males and females in terms of general attitude (male: 3.22, female: 2.82, $p < 0.005$) and perceived benefits (male: 3.60, female: 3.44, $p < 0.001$) towards GMOs. The mean values of these variables were higher for males than for females, suggesting that females have a more negative attitude towards GMOs. There are also significant differences in general attitude among different education fields. According to the ANOVA analysis, those who major in science have a more positive general attitude towards GMOs compared to those who major in humanities and social science ($p < 0.001$).

On the correlation analysis among the variables, We've conducted Pearson correlation analysis shows that subjective knowledge and objective knowledge is positively correlated with both perceived benefits and perceived risks ($p < 0.001$); objective knowledge is positively correlated with perceived benefits ($p < 0.001$); and perceived risks are positively correlated with perceived benefits ($p < 0.001$). This suggests that individuals with higher knowledge scores are more likely to recognize both the benefits and risks of GMOs. Moreover, individuals who perceive risks associated with GMOs also tend to recognize their potential benefits.

As for public opinion and the relationship between GMOs and SDGs, the results show that the public's views on the association between GMOs and SDGs are significantly positively correlated with subjective knowledge, objective knowledge, general attitude, perceived risk, perceived benefits, and trust ($p < 0.001$). This suggests that individuals with more knowledge about GMOs, and a positive attitude towards GMOs are more inclined to believe that GMOs can positively impact the SDGs.

## 4.  Discussion

The research results indicate a widespread awareness of GMOs among the public in Taiwan. A significant majority (97%) have at least some familiarity with GMO-related terms, with only 3.3% being completely unaware. However, when delving deeper into self-rated understanding, the results reveal that while many are somewhat familiar with GMOs (65.6%), a smaller percentage possess a high level of knowledge (1.5% very familiar). This suggests a superficial awareness that does not necessarily translate into comprehensive understanding. Objective knowledge assessments further highlight these gaps. For instance, only a minority correctly answered fundamental questions about genetic concepts, with just 14.7% knowing that genes are made up of chromosomes and 35.7% understanding that consuming GMO products cannot alter a consumer's genes. These findings align with global trends where public knowledge about GMOs often lags behind their prevalence in media and policy discussions. Studies in China, Europe and the United States have similarly shown that while the public is aware of GMOs, detailed understanding is often lacking [28,29]. The discrepancy between subjective and objective knowledge points to a need for more effective educational initiatives. Such initiatives should aim not only to increase awareness but also to deepen understanding of genetic science and biotechnology. This could involve integrating more detailed GMO content into school curricula and public information campaigns to address misconceptions

and build a more informed populace. Previous studies have emphasized the importance of education in improving public understanding of GMOs. For example, Zhang et al. [12] found that targeted educational programs significantly improved knowledge and reduced misconceptions about GMOs among students. Furthermore, the significant difference in knowledge levels across age groups, with younger individuals demonstrating higher subjective and objective knowledge, suggests that recent educational efforts may be more effective. However, the challenge remains to extend these gains across all demographic segments to ensure a well-rounded understanding of GMOs in the broader public. This trend is consistent with findings from other regions, where younger populations tend to be better informed about GMOs due to more recent exposure to relevant educational content [30,31]. In summary, while awareness of GMOs is high in Taiwan, there is a clear need to improve the depth of understanding. Bridging the gap between awareness and knowledge is crucial for informed public discourse and decision-making regarding GMO use and regulation. Enhanced education and targeted information dissemination are key strategies to achieve this goal, ensuring that the public is better equipped to engage with GMO-related issues critically and knowledgeably.

The public's attitudes towards GMOs in Taiwan present a nuanced picture, characterized by a blend of neutrality and moderate concern. The average attitude score of 3.02 reflects a generally neutral stance, but deeper analysis reveals varying levels of agreement and disagreement with specific statements. For example, while a significant portion of respondents (46.4%) held neutral views on whether GMOs violate the laws of nature, a notable portion of respondents (42.3%) agreed or strongly agreed with this statement, indicating underlying ethical and moral concerns. This complexity aligns with previous studies that suggest public attitudes towards GMOs are often ambivalent and context-dependent [32]. Moreover, when comparing our findings with another study conducted in East Asia [28], we observed that both studies identify a substantial portion of the population holding neutral attitudes. However, the Chinese study reveals a significantly higher level of opposition compared to Taiwan, where attitudes are more moderate and less polarized. The stark difference in negative perceptions in China may be influenced by the broader context of food safety scandals, which have eroded public trust in the government and increased scrutiny [33].

In terms of perceived risks, the results show that the public in Taiwan holds slightly below average risk perceptions (mean = 2.78), and concerns about unknown side effects and potential negative impacts on health and the natural ecosystem are not prevalent. Interestingly, both subjective and objective knowledge are positively correlated with perceived risks. Our findings are inconsistent with related research [34,35]. For instance, Purbosari & Ma'rifah [35] suggested that a higher level of knowledge regarding GMOs can lead to lower risk perception towards GMOs. One possible explanation is that the type of knowledge acquired by the public in Taiwan may be more factual than analytical, resulting in heightened awareness of potential risks without a deeper understanding of the scientific consensus. Additionally, media portrayals of GMOs in Taiwan often emphasize both benefits and controversies, which may reinforce this dual perception [10]. Cultural attitudes—such as skepticism toward government regulation and scientific authority—may also play a role [11,12]. These factors could contribute to a more nuanced or ambivalent interpretation of GMO-related information. Future research could explore how different types and sources of knowledge influence risk perception across various sociocultural contexts.

The public's perception of the benefits of GMOs is positive, with an average mean score of 3.52. Most respondents recognize the potential for GMOs to increase global food production (mean = 3.74) and stabilize food prices (mean = 3.55). These perceived benefits align with the broader scientific consensus on the advantages of GMOs in addressing food security and agricultural sustainability [36]. Our study found that objective knowledge is positively correlated with perceived benefits, which corresponds with previous studies [37], indicating that scientific information on GMOs can enhance perceived benefits and attitudes.

Moreover, our study found a significant lack of trust among the public in Taiwan regarding the management of GMOs. The overall mean trust score was 2.56, suggesting that respondents are generally skeptical about the ability of technology and the government to effectively regulate GMOs. Specifically, only 20.8% agree that GMOs can be properly regulated by technology, and just 11.9% have confidence in the government's management of GMOs. These low levels of trust contrast

with previous studies, which found relatively higher public trust in GMO management [38]. One possible explanation is that Taiwan has historically relied on top-down decision-making and positivistic approaches to risk assessment, which may have amplified social concerns and weakened institutional credibility [39]. Additionally, persistent cultural skepticism toward governmental and scientific institutions may further contribute to public doubt [11,23]. These contextual factors may help explain the low levels of trust observed in this study and underscore the need for more inclusive and transparent approaches to public engagement on GMO policy. This finding also suggests that the Taiwanese government may need to strengthen its efforts to communicate more effectively with the public about GMOs.

As to the demographic variables influencing public opinions on GMOs, gender differences play a role in shaping attitudes and perceptions. The present study found that males generally have a more positive attitude towards GMOs and perceive greater benefits compared to females. This gender disparity is supported by previous research, which suggests that men and women often differ in their risk perceptions and trust in scientific advancements [40]. Educational background further influences attitudes, with individuals majoring in science exhibiting more positive attitudes towards GMOs than those in humanities and social sciences, which corresponds with previous studies [41]. This trend suggests that scientific literacy may play a crucial role in shaping more favorable perceptions of GMOs.

Regarding the public's WTP, a premium for non-GMO and organic food products in Taiwan, the results show a notable inclination towards paying higher prices for organic foods compared to non-GMO foods, with a weighted mean WTP of 30.12% for organic food versus 25.4% for non-GMO food. This indicates a higher perceived value in organic products, possibly due to the extensive promotion of organic agriculture as being more natural and sustainable [42]. Moreover, the highest proportion of respondents (24%) were willing to pay an additional 11–20% for non-GMO food, followed closely by those willing to pay an extra 1–10% (20.2%). Previous research found that Taiwanese people were willing to pay approximately 19% more to purchase non-GMO food [43]. Compared to the past, the willingness to pay has increased by about 6%, indicating that people are now willing to spend more for non-GMO food. For organic food, the most common WTP was also 11–20% (22%), but a significant number (21.2%) were willing to pay an extra 21–30%. These findings suggest a strong market potential for organic products, driven by consumer perceptions of health benefits and environmental sustainability [44].

The public perception in Taiwan reflects a significant recognition of the positive impacts GMOs can have on certain SDGs. Most notably, a majority believe there is a positive relationship between GMOs and SDG 2, highlighting GMOs' potential to enhance food security by increasing agricultural productivity and reducing hunger. This aligns with existing literature suggesting GMOs play a crucial role in ensuring food security [36,45]. Additionally, many respondents perceive a positive relationship between GMOs and SDG 1, implying that GMOs can alleviate poverty by boosting crop yields and providing economic benefits to farmers, especially in developing regions [46]. A significant portion also sees GMOs positively impacting SDG 9, underscoring the role of biotechnology in driving agricultural innovation and supporting infrastructure development [47]. However, there are concerns regarding the negative impacts of GMOs on environmental sustainability. Many respondents believe GMOs negatively affect SDG 15, SDG 14, and SDG 12. These concerns reflect potential ecological risks associated with GMO cultivation, such as biodiversity loss and unintended consequences on ecosystems [48,49]. Interestingly, perceptions of GMOs' impact on SDG 5 are mixed, with a minority seeing negative effects and a smaller portion recognizing positive benefits. The majority hold a neutral view, indicating uncertainty or a perceived lack of direct relevance between GMOs and gender equality. This suggests a need for more gender-inclusive approaches in the deployment and management of GMOs to ensure equitable benefits across all societal groups [50]. The results show that public views on the association between GMOs and SDGs are significantly positively correlated with subjective knowledge, objective knowledge, general attitude, perceived risk, perceived benefits, and trust. This indicates that individuals with more knowledge about GMOs and a positive attitude towards them are more likely to believe that GMOs can positively impact the SDGs. Therefore, enhancing public education and communication about GMOs could improve their acceptance and perceived contribution to sustainable development.

## 5. Conclusion

This study provides a comprehensive assessment of public awareness, understanding, and attitudes towards GMOs in Taiwan, offering valuable insights and implications for policymakers and educators. The findings highlight significant gaps in public knowledge about GMOs, revealing that while awareness is high, deep understanding is lacking. The study also provides nuanced insights into public attitudes, perceived risks, and benefits of GMOs, emphasizing the importance of trust-building efforts by government and regulatory bodies. Additionally, it identifies strong market potential for organic and non-GMO products, driven by health and environmental considerations.

This study has two primary limitations. The first concerns the use of general biology and genetics questions to assess public knowledge. While these items were intentionally included to evaluate whether participants' perspectives on GMOs are grounded in fundamental scientific understanding or shaped by subjective impressions, they may not adequately capture more nuanced knowledge specific to GMO technologies. Future research could address this by incorporating targeted questions that differentiate perceptions of plant-based versus animal-based GMOs. The second limitation involves potential selection bias resulting from the sampling methods employed. The questionnaire was distributed online and through personal networks, which may have led to the overrepresentation of certain demographic groups, such as younger or more highly educated individuals. Although efforts were made to diversify outreach, the findings may not fully reflect the views of the broader population in Taiwan. Future studies could mitigate this issue by using stratified random sampling to improve representativeness.

Despite this, from the investigation of the relationship between GMOs and SDGs, the study explores the perceived benefits and risks of GMOs. It underscores the need for more effective educational initiatives and tailored communication strategies to address ethical concerns and enhance scientific literacy. These efforts are crucial for informed public discourse and decision-making regarding GMOs and for advancing the SDGs. These findings also carry important implications for policymakers. First, the significant gaps in public knowledge highlight the need for targeted educational initiatives that not only raise awareness but also deepen scientific understanding of GMOs. Second, the observed distrust in regulatory systems suggests that greater transparency and participatory approaches in GMO policy-making could be effective in rebuilding public confidence. By addressing both knowledge deficits and institutional skepticism, policymakers can help foster a more informed and engaged public, which is essential for the successful implementation of GMO-related technologies within a sustainable development framework.

## Supporting information

**S1 File.** Genetically modified organisms and sustainable development goals: a survey of Taiwanese public opinion. Genetically modified organisms and sustainable development questionnaire.
(PDF)

## Author contributions

**Conceptualization:** Shin-Cheng Yeh.

**Data curation:** Chih-Cheng Lin.

**Formal analysis:** Pei-Xuan Chen.

**Investigation:** Pei-Xuan Chen.

**Supervision:** Shin-Cheng Yeh.

**Visualization:** I-Chun Chiang.

**Writing – original draft:** Pei-Xuan Chen, Chih-Cheng Lin.

**Writing – review & editing:** Shin-Cheng Yeh.

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
