## [Decision Letter · Decision Letter 0]

PONE-D-24-50672Genetically Modified Organisms and Sustainable Development Goals: A Survey of Taiwanese Public OpinionPLOS ONE

Dear Dr. Lin,

Thank you for submitting your manuscript to PLOS ONE. After careful consideration, we feel that it has merit but does not fully meet PLOS ONE’s publication criteria as it currently stands. Therefore, we invite you to submit a revised version of the manuscript that addresses the points raised during the review process.

**Dear Authors,****I have gone through your manuscript as well as reviewer's comments. This is well written article. Reviewer's have raised some important concerns. You are requested too addressed all those and incorporate them in the revised version of your manuscript.** Please submit your revised manuscript by May 19 2025 11:59PM. If you will need more time than this to complete your revisions, please reply to this message or contact the journal office at plosone@plos.org . Please include the following items when submitting your revised manuscript:

We look forward to receiving your revised manuscript.

Kind regards,

Faiz Ahmad Joyia, Ph.D.

Academic Editor

PLOS ONE

2. We noticed you have cited a previous publication which has been retracted:

- Gaskell, G., Bauer, M. W., Durant, J., & Allum, N. C. (1999). Worlds apart? The reception of genetically modified foods in Europe and the US. Science, 285(5426), 384-387.

In your revision, please ensure you review this citation and any statements it supports. Please either remove the citation or provide an explanation for why it should be included. Further consideration is dependent on these concerns being addressed.

Additional Editor Comments:

Dear Author,

Please go through the reviewer's comments and incorporate in the revised version of manuscript.

Reviewers' comments:

Reviewer's Responses to Questions

**Comments to the Author**

1. Is the manuscript technically sound, and do the data support the conclusions?

Reviewer #1: Yes

Reviewer #2: Yes

2. Has the statistical analysis been performed appropriately and rigorously? 

Reviewer #1: Yes

Reviewer #2: Yes

3. Have the authors made all data underlying the findings in their manuscript fully available?

Reviewer #1: Yes

Reviewer #2: Yes

4. Is the manuscript presented in an intelligible fashion and written in standard English?

Reviewer #1: Yes

Reviewer #2: Yes

5. Review Comments to the Author

Reviewer #1: Overall, this is a well-written and informative article that addresses the public perception of GMOs in Taiwan. I recommend acceptance with minor revisions to improve clarity and strengthen the presentation of methods and results.

Methods Section (Section 2.1.2):

Section 2.1.2 contains a full paragraph of discussion, beginning with “Previous research has indicated”. This content is more suited for the Discussion section rather than the Methods section. I recommend relocating this paragraph to the Discussion section or removing it altogether to maintain the methodological focus of this section.

Similarly, other subsections within the Methods section frequently reference prior studies and their findings. These references should be moved to the Discussion section or removed to keep the Methods section focused solely on the study's design and execution.

Survey Design (Section 3.1):

The questions used to assess public knowledge on GMOs are somewhat broad and not specific enough to the subject matter. Several questions pertain to general biology and genetics rather than focusing directly on genetically modified organisms.

The survey design could be improved by including more targeted questions, such as those addressing public perceptions of GMO plant-based foods versus GMO animals. This would provide more nuanced insights into public opinion on GMOs.

Clarification in Results (Section 3.2):

Please clarify the statement “scored in reverse” in Section 3.2. It would be helpful to briefly explain the scoring method and the rationale for reverse scoring to ensure the reader understands how the data were interpreted.

The article presents valuable insights into public perception of GMOs in Taiwan, and with these minor adjustments, it will be even more impactful.

Reviewer #2: Dear editor, I reviewed the manuscript on “Genetically Modified Organisms and Sustainable Development Goals: A Survey of Taiwanese Public Opinion." I think the topic is very relevant as there is a global concern regarding GMOs. This study addresses the critical aspect of GMOs awareness and sustainable development goals. However, there are certain minor revisions which are required to be addressed before its publication.

1- In the introduction part the paragraphs pertaining to use of GMOs can be replaced with their public concerns and policies.

2- I could not find the research gap in the introduction part. What is the novelty of this study, which has not been addressed earlier?

3- In the material and methods section authors prepared several questionaries. However, authors did not point out any selection biasedness during surveys.

4- The participants consent for this study to be published is missing.

5- The results indicate that greater knowledge correlates with both higher perceived benefits and risks. However, this is inconsistent with some previous studies. Authors should highlight that why this discrepancy exists (e.g., cultural factors, education systems, media influence).

6- Authors found low public trust in GMO regulations. But they did not explain why this distrust exists?

7- I found that there is a missing part for policy makers? How can policymakers use these findings to improve public education and regulatory frameworks?

6. PLOS authors have the option to publish the peer review history of their article (what does this mean? ). If published, this will include your full peer review and any attached files.

**Do you want your identity to be public for this peer review?** For information about this choice, including consent withdrawal, please see our Privacy Policy .

Reviewer #1: No

Reviewer #2: **Yes: ** Muhammad Shah Nawaz-ul-Rehman

---

## [Author Response · Author response to Decision Letter 1]

24 Apr 2025

Responses to the Reviewers’ Comments

We are grateful for the reviewers' valuable and insightful comments, which have significantly contributed to the improvement of this study. In response, we have made concerted efforts to revise and supplement the manuscript accordingly. For your convenience, all changes have been highlighted in red. We hope that these revisions meet your expectations.

Reviewer #1:

1. Overall, this is a well-written and informative article that addresses the public perception of GMOs in Taiwan. I recommend acceptance with minor revisions to improve clarity and strengthen the presentation of methods and results.

Response: We sincerely thank you for your encouraging feedback. We have carefully reviewed the manuscript and made several clarifications and refinements throughout the Methods and Results sections to further enhance clarity and presentation. These changes are highlighted in the revised manuscript.

2. Section 2.1.2 contains a full paragraph of discussion, beginning with “Previous research has indicated”. This content is more suited for the Discussion section rather than the Methods section. I recommend relocating this paragraph to the Discussion section or removing it altogether to maintain the methodological focus of this section. Similarly, other subsections within the Methods section frequently reference prior studies and their findings. These references should be moved to the Discussion section or removed to keep the Methods section focused solely on the study's design and execution.

Response: We appreciate this helpful suggestion. In response, we have revised Sections 2.1.2, 2.1.3, and 2.1.4 by removing interpretative commentary and retaining only citations that directly inform the questionnaire design. We believe these revisions enhance both the clarity and structural coherence of the manuscript. The revised sections, located on page 5, are presented below.

2.1.2 Questionnaire design for the attitude dimension

The attitude dimension of the questionnaire, drawing on the findings of Scott et al. [21], includes four sub-dimensions: general attitude, perceived risks, perceived benefits, and trust. The scales in this dimension mostly employ a five-point Likert scale, ranging from strongly disagree to strongly agree, to evaluate public attitudes towards GMOs.

To construct the perceived risks scale, items were informed by common concerns identified in previous studies, including health, environmental, and socio-economic factors [5]. Items in the perceived benefits scale reflected findings that GMOs may increase crop yields, reduce production costs, and improve food quality [22]. Trust-related items were developed based on the work of Scott et al. [21], and Zhang et al. [2023], focusing on trust in government and scientific sources.

2.1.3 Questionnaire design for the WTP

WTP refers to the maximum price individuals are willing to pay for a product or service, reflecting the value they place on the item [24]. This study measured willingness to pay a premium for non-GMO and organic products. The questionnaire design was informed by previous research [25, 26], with responses categorized into percentage-based intervals (e.g., 0%, 1%–10%, 11%–20%, etc.) to estimate WTP.

2.1.4 Questionnaire design for the relationship between GMOs and SDGs

This section of the questionnaire assessed public perceptions regarding the relationship between GMOs and the SDGs. Respondents were first asked about their level of understanding of the SDGs and whether they had encountered any related images. We also assessed respondents’ self-perceived knowledge of the SDGs to better elucidate their basic understanding. The questionnaire then presented the 17 SDGs, accompanied by a five-point scale with the following options: very negative (−2), negative (−1), neutral (0), positive (+1), and very positive (+2). This scale enabled respondents to indicate their perceptions of the impact of GMOs on each of the 17 SDGs.

3. Survey Design (Section 3.1): The questions used to assess public knowledge on GMOs are somewhat broad and not specific enough to the subject matter. Several questions pertain to general biology and genetics rather than focusing directly on genetically modified organisms. The survey design could be improved by including more targeted questions, such as those addressing public perceptions of GMO plant-based foods versus GMO animals. This would provide more nuanced insights into public opinion on GMOs.

Response: Thank you for this thoughtful suggestion. We agree that including more specific items related to GMO applications (e.g., plant- vs. animal-based) could yield richer insights. In this study, we intentionally incorporated broader biology and genetics items to evaluate whether participants’ perspectives on GMOs were rooted in fundamental scientific understanding or influenced by subjective impressions. We believe this approach provided valuable context for interpreting public knowledge levels. Nevertheless, we have acknowledged this design choice as a limitation and proposed it as a direction for future research in the revised manuscript (p. 20, Conclusion section), as presented below.

This study has two primary limitations. The first concerns the use of general biology and genetics questions to assess public knowledge. While these items were intentionally included to evaluate whether participants’ perspectives on GMOs are grounded in fundamental scientific understanding or shaped by subjective impressions, they may not adequately capture more nuanced knowledge specific to GMO technologies. Future research could address this by incorporating targeted questions that differentiate perceptions of plant-based versus animal-based GMOs.

4. Clarification in Results (Section 3.2):

Please clarify the statement “scored in reverse” in Section 3.2. It would be helpful to briefly explain the scoring method and the rationale for reverse scoring to ensure the reader understands how the data were interpreted.

Response: We appreciate your helpful comment. To enhance clarity, we have revised Section 3.2—particularly the descriptions of Table 3 and Table 6—to provide a more detailed explanation of the reverse-scoring process. The revised content can be found on p. 8 and p.10, as presented below.

Table 3 presents the general attitude towards GMOs. Each level of agreement was scored from 1 (strongly disagree) to 5 (strongly agree). For negatively worded statements—such as “GMOs violate the laws of nature,” “GMOs violate moral principles,” and “GMOs violate religious beliefs”—responses were reverse-coded. This means that a response of 5 (strongly agree) was recoded as 1, and 1 (strongly disagree) as 5. Reverse scoring was used to ensure that across all items, higher scores consistently reflected a more favorable attitude toward GMOs.

Table 6 presents the trust of respondents towards GMOs. Each level of agreement was scored from 1 to 5 for the statements “GMOs can be properly regulated by technology” and “I have confidence in the government's management of GMOs.” Two negatively worded items statements “The development of GMO technology will exceed human control” and “The public should participate in the management of GMOs” were reverse-coded.

Reviewer #2:

1. This study addresses the critical aspect of GMOs awareness and sustainable development goals. However, there are certain minor revisions which are required to be addressed before its publication.

Response: We sincerely thank you for recognizing the significance of our study. We have carefully considered all of the suggested revisions and have made the necessary updates to enhance the clarity, structure, and academic rigor of the manuscript.

2. In the introduction part the paragraphs pertaining to use of GMOs can be replaced with their public concerns and policies.

Response: Thank you for this insightful suggestion. In response, we have revised the Introduction to present a more balanced perspective by incorporating public concerns and skepticism related to GMOs. The revised content can be found on page 2, as presented below.

Despite the potential benefits of GMOs, their development and deployment remain contentious. Public apprehension frequently centers on perceived health risks, environmental repercussions, and ethical concerns—issues often exacerbated by misinformation and media sensationalism [10]. In Taiwan, as in many other regions, persistent skepticism regarding GMO safety and a prevailing distrust in regulatory authorities continue to influence public attitudes [11, 12]. These concerns underscore the importance of examining not only the scientific and economic dimensions of GMOs but also the sociocultural and regulatory contexts that shape public acceptance. As Nasser et al. [13] emphasize, the adoption of GM technology poses several challenges that may hinder progress toward achieving the SDGs.

3. I could not find the research gap in the introduction part. What is the novelty of this study, which has not been addressed earlier?

Response: Thank you for your comment. To address your point, we have revised the final paragraph of the Introduction to more explicitly highlight the novelty and contribution of our study. Specifically, we emphasize the limited research on public perceptions of GMOs and the SDGs in East Asia and clarify how our study addresses this gap by providing a comprehensive analysis focused on Taiwan. The revised paragraph is presented below on page 3.

It is essential to identify current attitudes toward biotechnology in different countries to understand public fears and determine potential knowledge gaps [19]. Taiwan, as a developed country in East Asia, has shown through past research that its citizens support the development of GM technology because it is a global trend [20]. However, Chen [11] found that the public's willingness to consume GM foods was not high, with a mean score of 3.16 out of 7. Therefore, there is a need to investigate public opinion regarding GMOs in Taiwan. Moreover, while public perceptions of GMOs have been extensively studied in Western contexts—particularly concerning the preference for “natural” food and the perceived unnaturalness of GMOs [21]—there is limited research comparing willingness to pay (WTP) for GM, non-GM, and organic food in the East Asian context. Likewise, although GMOs have been framed as contributors to the SDGs in prior literature [7, 8], few studies have explored public perceptions linking GMOs and SDGs in East Asia The research purposes of this study include: first, investigating the public's knowledge, attitudes, and WTP concerning GMOs. Second, we analyze the public's perceived relationship between GMOs and the SDGs. This study offers valuable insights into the complex dynamics of public opinion on GMOs and their potential impact on sustainable development initiatives. By focusing on the Taiwanese context, we provide new empirical evidence on how GMOs are perceived in relation to sustainable development, addressing both regional and thematic gaps in the existing literature.

4. In the material and methods section authors prepared several questionnaires. However, authors did not point out any selection biasedness during surveys.

Response: Thank you for raising this important point. We have addressed the potential for selection bias in the revised manuscript by adding a brief discussion in the Limitations section. While we aimed to achieve broad distribution through online and network-based methods, we acknowledge that the sample may not fully represent all demographic segments. This limitation has been explicitly noted and highlighted as an area for improvement in future research. The revised text is presented below on page 20.

The second limitation involves potential selection bias resulting from the sampling methods employed. The questionnaire was distributed online and through personal networks, which may have led to the overrepresentation of certain demographic groups, such as younger or more highly educated individuals. Although efforts were made to diversify outreach, the findings may not fully reflect the views of the broader population in Taiwan. Future studies could mitigate this issue by using stratified random sampling to improve representativeness.

5. The participants consent for this study to be published is missing.

Response: Thank you for your comment. We have added a clarification in Section 2.2 indicating that informed consent was obtained through the Google Form. A statement outlining the purpose of the study and the use of anonymized data was presented at the beginning of the form. By submitting their responses, participants provided consent for their data to be used in academic publications. The revised text is presented below on p. 6.

In addition, a consent statement was included on the introduction page of the Google Form, explaining the purpose of the study, data anonymization, and the intended use of results for academic publication. Proceeding to complete the survey was taken as confirmation of participants’ consent to participate and for their anonymized responses to be used in this study

6. The results indicate that greater knowledge correlates with both higher perceived benefits and risks. However, this is inconsistent with some previous studies. Authors should highlight that why this discrepancy exists (e.g., cultural factors, education systems, media influence).

Response: Thank you for this insightful comment. In response, we have expanded the Discussion section to more explicitly acknowledge the inconsistency between our findings and previous studies regarding the relationship between knowledge and perceived risk. Please see the revised text on p.17, presented below.

One possible explanation is that the type of knowledge acquired by the public in Taiwan may be more factual than analytical, resulting in heightened awareness of potential risks without a deeper understanding of the scientific consensus. Additionally, media portrayals of GMOs in Taiwan often emphasize both benefits and controversies, which may reinforce this dual perception [10]. Cultural attitudes—such as skepticism toward government regulation and scientific authority—may also play a role [11, 12]. These factors could contribute to a more nuanced or ambivalent interpretation of GMO-related information. Future research could explore how different types and sources of knowledge influence risk perception across various sociocultural contexts.

7. Authors found low public trust in GMO regulations. But they did not explain why this distrust exists?

Response: hank you for this insightful comment. In response, we have revised the Discussion section to provide a more thorough explanation of the low levels of public trust observed in our study. The revised text is presented below on p.18.

One possible explanation is that Taiwan has historically relied on top-down decision-making and positivistic approaches to risk assessment, which may have amplified social concerns and weakened institutional credibility [39]. Additionally, persistent cultural skepticism toward governmental and scientific institutions may further contribute to public doubt [11, 23]. These contextual factors may help explain the low levels of trust observed in this study and underscore the need for more inclusive and transparent approaches to public engagement on GMO policy. This finding also suggests that the Taiwanese government may need to strengthen its efforts to communicate more effectively with the public about GMOs.

8. I found that there is a missing part for policy makers? How can policymakers use these findings to improve public education and regulatory frameworks?

Response: Thank you for pointing out this important aspect. We have revised the Conclusion section to explicitly address the implications of our findings for policymakers. In particular, we emphasize that enhancing public education, increasing transparency, and promoting participation in regulatory processes may help build trust and foster informed public engagement with GMO technologies. The revised text is presented below on p.21.

These findings also carry important implications for policymakers. First, the significant gaps in public knowledge highlight the need for target

---

## [Editor Report · Decision Letter 1]

Genetically Modified Organisms and Sustainable Development Goals: A Survey of Taiwanese Public Opinion

PONE-D-24-50672R1

Dear Dr. Lin,

We’re pleased to inform you that your manuscript has been judged scientifically suitable for publication and will be formally accepted for publication once it meets all outstanding technical requirements.

Kind regards,

Faiz Ahmad Joyia, Ph.D.

Academic Editor

PLOS ONE
---

## [Editor Report · Acceptance letter]

PONE-D-24-50672R1

PLOS ONE

Dear Dr. Lin,

I'm pleased to inform you that your manuscript has been deemed suitable for publication in PLOS ONE. Congratulations! Your manuscript is now being handed over to our production team.

Kind regards,

on behalf of

Dr. Faiz Ahmad Joyia

Academic Editor

PLOS ONE